# Food Waste-Assisted Metal Extraction from Printed Circuit Boards: The *Aspergillus niger* Route

**DOI:** 10.3390/microorganisms9050895

**Published:** 2021-04-22

**Authors:** Carlotta Alias, Daniela Bulgari, Fabjola Bilo, Laura Borgese, Alessandra Gianoncelli, Giovanni Ribaudo, Emanuela Gobbi, Ivano Alessandri

**Affiliations:** 1B+LabNet-Environmental Sustainability Lab, University of Brescia, Via Branze 45, 25123 Brescia, Italy; carlotta.alias@unibs.it; 2Agri-Food and Environmental Microbiology Platform (PiMiAA), Department of Molecular and Translational Medicine, University of Brescia, Via Branze 45, 25123 Brescia, Italy; emanuela.gobbi@unibs.it; 3Chemistry for Technologies Laboratory, Department of Mechanical and Industrial Engineering, University of Brescia, Via Branze 38, 25123 Brescia, Italy; fabjola.bilo@unibs.it (F.B.); laura.borgese@unibs.it (L.B.); 4Piattaforma di Proteomica, AgroFood Lab, Department of Molecular and Translational Medicine, University of Brescia, Viale Europa 11, 25123 Brescia, Italy; alessandra.gianoncelli@unibs.it (A.G.); giovanni.ribaudo@unibs.it (G.R.); 5Department of Information Engineering, University of Brescia, Via Branze 38, 25123 Brescia, Italy; ivano.alessandri@unibs.it; 6Consorzio Interuniversitario Nazionale per la Scienza e Tecnologia dei Materiali, UdR Brescia, Via Branze 38, 25123 Brescia, Italy; 7Istituto Nazionale di Ottica—INO-CNR, UdR Brescia, Via Branze 38, 25123 Brescia, Italy

**Keywords:** fungus, acid solution, organic waste, electronic waste, solid state fermentation, circular economy, valuable metals

## Abstract

A low-energy paradigm was adopted for sustainable, affordable, and effective urban waste valorization. Here a new, eco-designed, solid-state fermentation process is presented to obtain some useful bio-products by recycling of different wastes. Urban food waste and scraps from trimmings were used as a substrate for the production of citric acid (CA) by solid state fermentation of *Aspergillus niger* NRRL 334, with a yield of 20.50 mg of CA per gram of substrate. The acid solution was used to extract metals from waste printed circuit boards (WPCBs), one of the most common electronic waste. The leaching activity of the biological solution is comparable to a commercial CA one. Sn and Fe were the most leached metals (404.09 and 67.99 mg/L, respectively), followed by Ni and Zn (4.55 and 1.92 mg/L) without any pre-treatments as usually performed. Commercial CA extracted Fe more efficiently than the organic one (123.46 vs. 67.99 mg/L); vice versa, biological organic CA recovered Ni better than commercial CA (4.55 vs. 1.54 mg/L). This is the first approach that allows the extraction of metals from WPCBs through CA produced by *A. niger* directly grown on waste material without any sugar supplement. This “green” process could be an alternative for the recovery of valuable metals such as Fe, Pb, and Ni from electronic waste.

^‡^ Co-last author.

## 1. Introduction

Electronic and food wastes are increasingly pressing issues that should be addressed in the near future. The rapid turnover of electronic devices such as personal computers, mobile phones, wearables, and smart home devices boosts the production of electrical and electronic equipment waste (WEEE) [1]. This stream is trending to reach more than 12 million tons by 2020 in Europe [2] and more than 52.2 million tons by 2021 worldwide [3]. Although printed circuit boards (PCBs) represent up to 6% of the WEEE mass, they have the most important environmental restrictions but, at the same time, the most economic advantages and are a secondary source for rare earth elements [4]. Price fluctuations, resource scarcity, availability, and access to concerns make it necessary to address the mining of secondary resources and reduce the demand for primary material [5].

Within the paradigm of a circular economy, urban mining of PCBs [6] could indeed become a source of secondary valuable raw materials such as the metals Cu, Zn, Pb, Ni, Cd, Au, Ag, and Sn [3,7,8]. Contrary to chemical, mechanical and pyrometallurgical processes, the use of microorganisms to solubilize metal oxides and sulfides (bioleaching) allows the recovering of metals from e-waste with environmentally friendly and cost-effective operations, reducing health concerns [9]. *Aspergillus niger* is a very versatile fungus that is able to solubilize metals from spent lithium ion mobile phone batteries and WPCBs, mainly due to the production of organic acids such as citric, oxalic, gluconic, and malic acid [10,11]. Interestingly, *A. niger* is able to grow on low-cost fermentation substrates of agro-industrial origin such as grape pomace, sugarcane bagasse, banana peel, or rice straw [12,13]. To date, only few studies have been conducted to take advantage of this route, which would allow cheap, low-energy citric acid (CA) production for industrial applications [14,15]. However, this option could open the way to the intriguing possibility of obtaining CA using food waste as a substrate. The same CA could be then directly exploited for recovering metals by WEEE leaching. In 2019, the FAO defined food waste as “the decrease in the quantity or quality of food resulting from decisions and actions by retailers, food services and consumers” [16]. Food waste production is estimated around 90 million tons per year in Europe, with an estimated cost of 143 billion Euros [17] and a global warming impact in the UK alone of between 2 and 3.6 ton CO_2_-eq. t^−1^ [18]. The problems related to these kinds of waste prompted the European Parliament to proclaim 2014 to be “the European Year against Food Waste” and to adopt a resolution that would lead to a 50% reduction in waste by 2025. In addition, the re-use of e-waste is an important aspect in the circular economy perspective, in which waste should not be considered as only “rubbish” but as a source of materials for new production cycles. This study aimed at developing a low-energy sustainable route for extracting metals from circuit boards, which was based on leaching operated by the organic acids (in particular, CA) produced by *A. niger*’s solid state fermentation (SSF) of food and garden waste. The experimental design of this approach is summarized in Figure 1.

## 2. Materials and Methods

### 2.1. Chemicals

Potato dextrose agar, methanol, citric acid, sodium hydroxide, phenolphthalein, and the gallium standard solution were all purchased from Sigma-Aldrich, St. Louis, MO, USA.

### 2.2. Substrate Preparation and Solid-State Fermentation

The non-profit local organization “Cauto” collected food waste (fruits and vegetables no longer suitable for sale) from large-scale organized distribution and scraps from yard trimmings. Food waste and scraps were separately dry milled in a blender until the particles reached 1–2 mm in diameter. The different components were weighed and mixed in micropropagation containers (Microbox, Micropoli, Cesano Boscone, Italy), as reported in Table 1. Filled boxes were subjected to 2 consecutive cycles of sterilization (121 °C for 15 min) and allowed to dry under a laminar flow hood for 16 h. *Aspergillus niger* NRRL 334 (Agricultural Research Service Culture Collection) was routinely grown on slants of potato dextrose agar at 26 °C. Two milliliters of conidia suspension (5 × 10^5^ conidia/mL in sterile water) was seeded in each box and incubated at 26 °C and 40% relative humidity (RH) under illumination with 16 h light/8 h dark cycles, using daylight tubes (24 W/m^2^, 9000 lx) in a climatic chamber (model 720, Binder, Tuttlingen, Germany). Similarly, non-fermented substrate was incubated as a control. The fungal growth was indirectly measured by daily visual observation of the substrates. The fermentation process was considered complete when the fungus entirely colonized the SSF substrate and was fully sporulating (7 days).

### 2.3. Citric Acid Production and Determination

After 7 days of culture, the substrates were submerged in 500 mL of distilled water and stirred for 15 min. The suspension was then filtered through 3 layers of sterile Miracloth (Sigma-Aldrich, St. Louis, MO, USA) to eliminate conidia and mycelial fragments. The flow-through was centrifuged at 6000× rpm and filtered using a membrane with a 0.45 µm pore size. The pH of the obtained solutions was measured with a pH meter (Hanna Instruments, Woonsocket, RI, USA) to assess the substrate’s acidification. Identification of CA was performed by electrospray ionization mass spectrometry (ESI-MS). Mass spectra were recorded by direct infusion ESI on a LCQ Fleet ion trap spectrometer (Thermo Fisher Scientific, Waltham, MA, USA). The instrument was set in negative ionization mode with a 5.0 kV spray voltage, 225 °C −1.0 V capillary voltage, and −40 V tube lens values. Gas flow rates (arb): sheath, 10; aux, 0; sweep, 0. The infusion flow rate was set to 5 mL/min. Samples were diluted in methanol (1:1000) before acquisition to obtain a stable ESI signal. CA concentration was then determined by titration with 0.1 M NaOH using phenolphthalein as an indicator. This experiment was performed in duplicate.

### 2.4. Metal Leaching

The wasted printed circuit boards (WPCBs) utilized in this study were obtained by disassembling out-of-use hand-held pointing (“mouse”) devices. The WPCBs were sterilized (121 °C for 15 min) to avoid contamination of the samples without any other pre-treatment such as removing plastic, electronic parts (PCI and chip slots), and chemical coating (solder mask). The whole WPCBs were then soaked in 50 mL of the CA solution obtained from SSF (CA 334). Solutions of commercial CA (CA Com) were utilized at the same concentration (10.67 mM) as a reference for leaching. Both samples were maintained at room temperature. An aliquot of each sample was collected after 1, 2, 4 and 7 days of exposure. All the experiments were performed in duplicate.

### 2.5. Determination of the Metal Content of Leachates by Total Reflection X-ray Fluorescence Analysis

Sample preparation for total reflection X-ray fluorescence (TXRF) analysis was performed following the validated procedure as described by Borgese et al. [19]. Briefly, sample solutions were prepared by weighing 90 µL of leachates and adding 10 µL of a Ga standard solution, which was used as internal standard for quantitative analysis, to obtain a final Ga concentration of 0.1 mg/L. From each sample, 3 replicates were prepared by depositing 10 µL of the specimen on the center of a siliconized quartz glass reflector and drying it on a hot plate at 50 °C. TXRF measurements were performed with the TXRF spectrometer S2 PICOFOX (Bruker AXS Microanalysis GmbH, Berlin, Germany) equipped with a Mo tube, operating at 50 kV and 750 μA; a multilayer monochromator (17.5 keV); FWHM 139.43 eV; and a silicon drift detector (SDD). The spectral fitting and deconvolution were performed by the SPECTRA instrument software after manual identification of the elements.

### 2.6. Statistical Analysis

The statistical analyses were performed by Microsoft Excel (2019). The statistical correlation between groups was analyzed by Student’s *t*-test. The *p*-values are indicated with asterisks: * *p* < 0.05; ** *p* < 0.01; *** *p* < 0.001. Differences were considered significant at *p*-values less than 0.05.

## 3. Results and Discussion

### 3.1. Citric Acid Production

The aqueous solution obtained from the SSF of *Aspergillus niger* NRRL 334 showed a pH reduction to 3.29, considerably lower than the pH value of the control solution (5.00). ESI-MS analysis revealed the presence of the CA peak (191.15 *m/z*) only in the SSF solution, confirming the ability of *A. niger* NRRL 334 to produce CA on organic waste (Figure 2B,C). Collision-induced dissociation (CID) experiments were performed to characterize the molecular ions detected at *m*/*z* = 191.15 (exact mass calculated for CA in negative ionization mode: 191.02, $C_{6}H_{7}O_{7}$^−^). The fragmentation was promoted by isolating the precursor ion in the trap and increasing the “normalized collision energy” parameter. The fragmentation pattern (*m*/*z* = 172.95, *m*/*z* = 111.03) was found to be in agreement with literature data [20,21] and with the spectrum recorded on a 10 mM commercial CA solution in methanol, thus confirming the identity of the compound (Figure 2D,E). On the other hand, the signal corresponding to CA was not detected in the control solution under the same ESI-MS experimental conditions (Figure 2A). The full-size ESI-MS spectra are shown in Appendix A.

*A. niger* is one of the most studied fungi for CA production from different substrates ranging from raw materials to standard culture media [13]. Until now, CA fermentation processes (surface-, submerged- or solid-state) using *A. niger* were carried out on a single type of raw material e.g., coconut cake or banana peel (among others) [22,23]. In this work, with the aim of food waste reuse, the SSF was performed on a mix of fruits and scraps from yard trimmings, in the absence of any other supplement. The chosen substrate was able to sustain both fungal growth and good production of CA. Moreover, the composition of this substrate was representative of the biodegradable municipal waste that usually contains food and green waste and that requires efficient circular utilization. The majority of municipal waste generated in Europe is still disposed of through landfilling (24%) or incineration (27%), with less than half recycled (31%) and composted (17%) [24]. Therefore, the present approach could help to divert fermentescible organic wastes from landfills and to guarantee that high-quality secondary raw materials are produced. The remaining waste could then be used for agricultural compost production [25] or as a fungal pre-treated feedstock for anaerobic digestion to produce renewable energy [26].

Due to the predominance of CA in the biological solution, a classic titration was carried out in order to determine the concentration of CA. A molarity of 10.67 mM and a concentration of 2.05 g/L were calculated. The obtained yield was 20.50 mg of CA per gram of substrate. CA production is influenced by the media composition, ranging from 19 to 766 g/kg when grown on a high sugar content (e.g., apple pomace or coconut cake) [13]. The amount of CA produced in this work from raw materials with a low sugar content is comparable with that obtained from a high-sugar substrate [27]. This result sheds new light on the possibility of directly using biodegradable waste for producing a high-value product such as CA, fostering the achievement of a food waste reduction milestone.

### 3.2. Metal Leaching

After disassembly of “mouse” devices, sterilized WPCBs were soaked in two different CA solutions, both at 10.67 mM. The following elements of interest were quantified: Br, Cr, Cu, Fe, Mn, Ni, Pb, Sn, Sr, and Zn (Table 2). Among these, the most abundant metal recovered was Sn, leached by CA 334 and CA Com solutions at concentrations above 400 mg/L (404.09 and 495.10 mg/L, respectively). Leachates were also strongly enriched in Fe (67.99 and 123.46 mg/L in CA 334 and CA Com, respectively) and Ni (4.55 and 1.54 mg/L in CA 334 and CA Com, respectively). Both the CA 334 and CA Com solutions increased the concentrations of Zn (1.92 and 1.75 mg/L), Br (0.62 and 0.65 mg/L), and Pb (0.12 and 0.16 mg/L). From these results, it can be inferred that the CA 334 produced by SSF leaches metals, with yields comparable with a commercial acid. Among the organic acids produced by *A. niger*, CA has an important role in bioleaching Cu, Mn, and Ni from WEEE [10,11]. On the other hand, oxalic acid is claimed to be the best leaching agent for Zn [10,28,29] according to our data. The metals solubilized by CA 334 from a WEEE could lead to (i) e-wastes with less environmental impact and/or (ii) a source for sustainable electronics. The acid solutions deteriorate the WPCB surface, allowing the leaching of some hazardous pollutants such flame retardants (Br) or heavy metals (Pb), as ruled by the European Directive on waste electrical and electronic equipment [30]. Removing brominated flame retardants added to the plastics used in WEEE require strong thermochemical treatments [31,32], not comparable with that presented in this study. However, despite a mild and short treatment, a significant rising trend in Br concentrations was detected in both the CA 334 and CA Com solutions (0.62 and 0.65 mg/L, respectively, *p* < 0.001). Moreover, the same trend was detected for Pb, with significant rising concentrations after 7 days (0.12 and 0.16 mg/L, respectively, *p* < 0.001).

These results were obtained without the application of physical processes (high temperatures or shredding) or chemical treatments (e.g., NaOH). Furthermore, the treatment of the whole board, instead of the standard pulverized form, was demonstrated to facilitate complete recycling of the non-metallic parts [33]. This experimental design could combine the reduction of input (energy and chemicals) and the potential complete recycling of the entire WPCB.

Interestingly, both CA solutions were able to leach metals in a time-dependent manner even if they started (T0) with a different metal composition (Table 2). In detail, Fe and Zn were detected in all the solutions less than 0.1 mg/L, while Cu was found only in the CA 334 solution at 0.013 mg/L. Sn, Fe, Mn, and Pb were leached by CA Com at higher concentrations but leaching activity was also exerted, with a similar trend, by CA 334 (Figure 3). In detail, slightly higher performance of CA Com was detected for Sn solubilization (Figure 3A). Moreover, CA Com permitted more efficient solubilization of Fe compared with CA 334 since the first day (37.00 vs. 19.8 mg/L, *p* < 0.01) to the last time point (123.46 vs. 67.99 mg/L, *p* < 0.001) (Figure 3B). On the other hand, Ni was more efficiently removed by CA 334 than CA Com, with significantly strong leaching activity after 4 (2.67 vs. 1.39 mg/L, *p* < 0.01) and 7 days (4.55 vs. 1.54 mg/L, *p* < 0.001) of contact (Figure 3C). The higher amount of Ni was achieved by the acid solution produced by *A. niger* NRLL 334, probably due to the presence of other acids derived from the Krebs cycle. It is well known that *A. niger* produces citric, oxalic, gluconic, and malic acid [10,11] during the fermentation process. As reported by Astuti and colleagues, citric acid allows the higher recovery of Ni from ores, but other organic acids produced by the Krebs cycle also leach Ni [34]. Further studies will be carried out to demonstrate the combined effects of Krebs organic acids on Ni extraction from WPCB. The CA Com solution was significantly more effective at leaching Mn (*p* < 0.001, Figure 3E) and Pb (*p* < 0.01, Figure 3F) compared with CA 334.

It is noteworthy to point out that a longer time is necessary to define the maximum recovered amounts of metals such as Cu. It was demonstrated that *A. niger* is capable of leaching high amounts of Zn, Ni, and Cu in 30 days from ground WPCBs [11]. Here, according to a low-energy paradigm, the WPCBs were processed without grinding, with a possible limitation of CA infiltration. Even though Ni and Cu are easily dissolved by CA [28], the small amount recovered in this study could be due to their unavailability to CA action due to epoxy coverage [35] or the absence of oxidant reagents in addition to the acid solution. H_2_O_2_ was added to an *Acidiphilium acidophilum* culture supernatant to recover a high amount of Cu from PCB [36]. Moreover, Jadhav and co-workers [33] reported that a fast and complete metal leaching from PCB was achieved using CA and 5.83% $H_{2}O_{2}$. Nevertheless, this preliminary study demonstrates the sustainability of this “green” process for the recovery of valuable metals such as Fe, Pb, and Ni from WEEE, broadening the horizons of food waste applications [37,38,39].

## 4. Conclusions

This study presents a low-energy, sustainable route for extracting metals from circuit boards, based on leaching by CA produced by *A. niger* under SSF of food and garden waste. The presented approach allows the reuse of organic waste (a mix of food waste and scraps from yard trimmings) as a substrate for the SSF of *A. niger*, in the absence of any other sugar supplement. *A. niger* secreted organic acids, particularly citric acid, which were able to leach metals such as Sn, Fe, Zn, and Ni from intact WPCBs without any energy-consuming treatments (WPCB grinding, high temperatures). Interestingly, the leaching activity of the biological organic acid solution was comparable with that of a CA commercial solution, suggesting a possible application of *A. niger* fermentation bio-products in green metal extraction from WPCBs.

This paves the way to a new approach to waste management in a circular economy view, where an organic waste assists the recovery and promotes the recycling of a different (inorganic, electronic) waste through a biological organism.

## Figures and Tables

**Figure 1 microorganisms-09-00895-f001:**
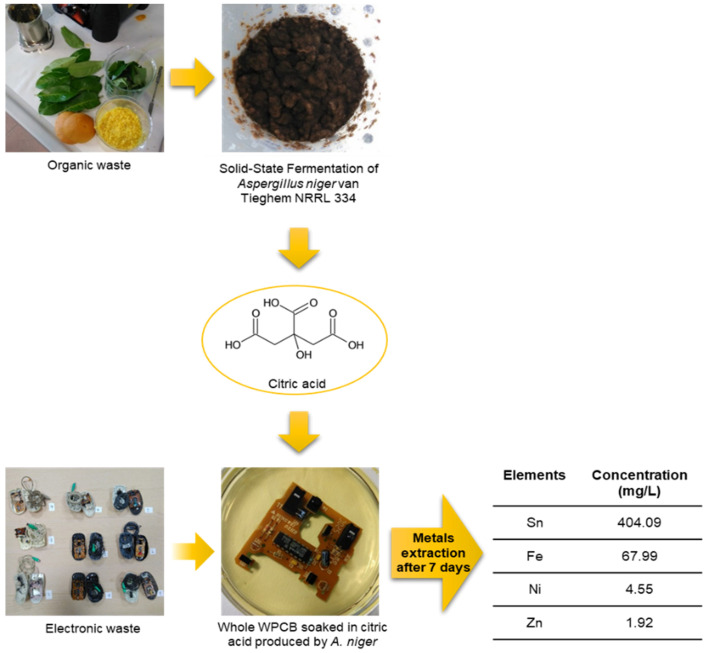
Schematic representation of the experimental design.

**Figure 2 microorganisms-09-00895-f002:**
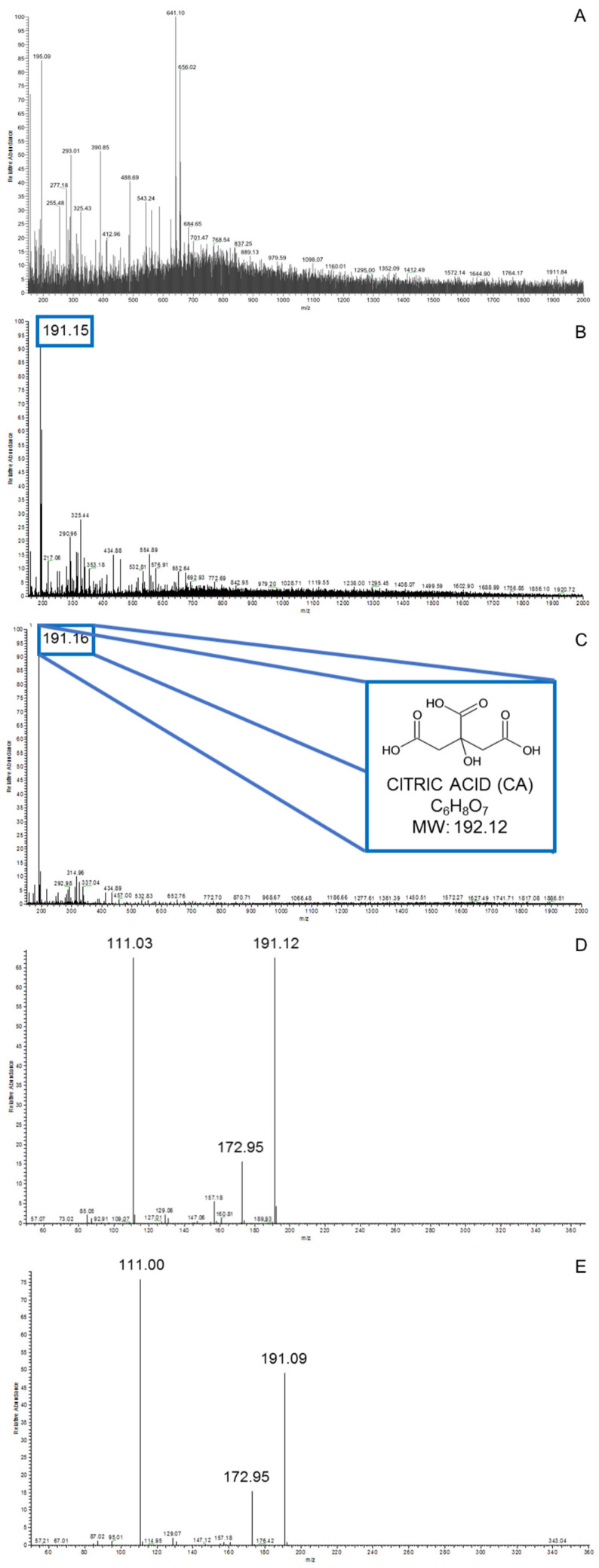
Negative ionization ESI-MS analysis of the non-fermented substrate extract (negative control) (**A**) and biological (**B**) and commercial (**C**) solutions of CA. Fragmentation pattern analysis (normalized collision energy = 20.00) validated the peak association to CA in the biological sample (**D**) compared with the commercial one (**E**).

**Figure 3 microorganisms-09-00895-f003:**
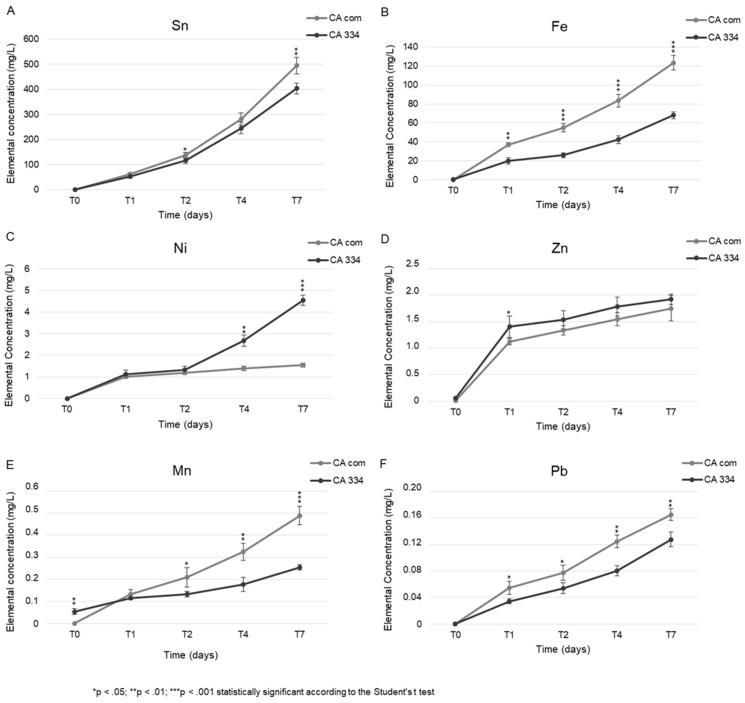
Trends of Sn (**A**), Fe (**B**), Ni (**C**), Zn (**D**), Mn (**E**), and Pb (**F**) amounts solubilized by the CA solution produced by *Aspergillus niger* NRRL 334 (CA 334) and by the commercial one (CA Com) over time, from T0 to 7 days (T7) of mouse circuit boards soaking. Data represent average concentrations from three independent measures ± uncertainty; * *p* < 0.05; ** *p* < 0.01; *** *p* < 0.001: statistically significant according to Student’s *t*-test.

**Table 1 microorganisms-09-00895-t001:** Components of the culture substrate for solid state fermentation of *Aspergillus niger.*

Constituents	% (*w/w*)	Amount per Culture (g)
Banana (peel and pulp)—*Musa acuminata* Cavendish Subgroup	30	15
Orange peels—*Citrus sinensis*	30	15
Aubergine—*Solanum melongena* L.	5	2.5
Courgette—*Cucurbita pepo* L.	5	2.5
Scraps from yard trimmings—*Prunus laurocerasus*	30	15

**Table 2 microorganisms-09-00895-t002:** Concentration (mg/L) of representative leachates generated by a citric acid (CA) solution produced by *Aspergillus niger* NRRL 334 (CA 334) and by a commercial CA (CA Com) (10.65 mM) at the beginning (T0) and after 7 days (T7) of mouse circuit boards soaking. Data represent average concentrations from three independent TXRF measures ± uncertainty.

Elements	CA 334		CA Com	
	T0	T7	T0	T7
Br	0.06 ± 0.01	0.62 ± 0.03 ***	<LOD	0.65 ± 0.09 **
Cr	<LOD	0.04 ± 0.02	<LOD	0.08 ± 0.05
Cu	0.013 ± 0.004	0.04 ± 0.01 *	<LOD	0.04 ± 0.03 *
Fe	0.09 ± 0.01	67.99 ± 3.52 ***	0.01 ± 0.00	123.46 ± 7.69 ***
Mn	0.05 ± 0.01	0.25 ± 0.01 ***	<LOD	0.49 ± 0.04 ***
Ni	<LOD	4.55 ± 0.24 ***	<LOD	1.54 ± 0.10 ***
Pb	<LOD	0.12 ± 0.01 ***	<LOD	0.16 ± 0.01 ***
Sn	<LOD	404.09 ± 21.15 ***	<LOD	495.10 ± 33.20 ***
Sr	0.031 ± 0.022	0.04 ± 0.01	<LOD	0.05 ± 0.01 **
Zn	0.06 ± 0.01	1.92 ± 0.10 ***	0.017 ± 0.006	1.75 ± 0.24 **

Note: * *p* < 0.05; ** *p* < 0.01; *** *p* < 0.001: statistically significant versus T0 values according to Student’s *t*-test. LOD: limit of detection.

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
