# Peer review of "Food Waste-Assisted Metal Extraction from Printed Circuit Boards: The Aspergillus niger Route"

_microorganisms, 2021, doi:10.3390/microorganisms9050895_

Round 1

Reviewer 1 Report

The manuscript describes good and interesting results regarding urban waste valorization, through solid-state fermentation with Aspergillus niger for citric acid production. The produced citric acid was furthermore subjected to metal extraction from printed circuit boards. The research subject is quite interesting, the study is well planned and executed. Although the solid-state fermentation should be better described, both in the MM and in the results section (how was the fermentation monitored? after SSF was the substrate analyzed? what kind of organic acids were produced and in what concentration?).

For further improvement, some corrections are required:

  • Line 22 “Here a combined recycling of different wastes is presented by a new,” please rephrase
  • line 28 if citric acid was abbreviated then ue the abbreviated form
  • line 29 “(4.55 and 1.92 mg/L, respectively)” please remove the second respectively
  • Line 40 – please correct “address” to “addressed”
  • line 62 – the abstract and the article should be considered separate texts, please define the abbreviation in the text also (CA).
  • line 64 – If CA is already abbreviated then use the abbreviated form after
  • line 89 – please put a space between the number and ℃
  • line 92 –? - the author wanted to specify the spore suspension adjustment? 106?
  • line 93 – RH – please define the abbreviation
  • Line 102 - Please introduce the country where the instruments (pH-meter, LCQ Fleet ion trap spectrometer, etc) were manufactured.
  • line 242, 258 - please correct to italics (in the whole manuscript).

Overall, the manuscript presents important experiments and results with significant data which is sufficiently described and discussed. After major improvements and corrections, the manuscript might be published.

Author Response

Point-by-point response to reviewer

Here, we answer point by point to comments raised by reviewer to the manuscript entitled "Food waste-assisted metal extraction from printed circuit boards: the Aspergillus niger’s route”

We have revised the manuscript in line with the following points raised by the Reviewer.

Comment: the solid-state fermentation should be better described, both in the MM and in the results section (how was the fermentation monitored? after SSF was the substrate analyzed? what kind of organic acids were produced and in what concentration?).

Answer: as suggested by the reviewer we added more information in the material and methods section:

Section 2.1: ‘The fungal growth was indirectly measured by daily visual observation of the substrates. The fermentation processes was considered complete when the fungus entirely colonized the SSF substrate and it was fully sporulating (7 days).

Section 2.2: ‘The suspension was then filtered through three layers of sterile Miracloth (Sigma-Aldrich, St. Louis, MO, USA) to eliminate conidia and mycelial fragments.’

It is well known that A. niger produces citric, oxalic, gluconic and malic acid during fermentation process. In our study the peak associated with CA is the dominant one as shown ESI-MS full spectra. Other krebs acids could be produced in SSF but, on the basis of our results, we can speculate that they are negligible in comparison to CA. This assumption was also confirmed by the leaching ability of the SSF extract (CA334), mainly composed by CA, that was comparable to the commercial CA.

For further improvement, some corrections are required:

Comment: Line 22 “Here a combined recycling of different wastes is presented by a new,” please rephrase

Answer: The sentence was rephrased and now reads as ‘Here a new, eco-designed, solid-state fermentation process is presented to obtain some useful bio-products by recycling of different wastes’

Comment: line 28 if citric acid was abbreviated then use the abbreviated form

Answer: Citric acid was replaced through out the text with the abbreviation CA

Comment: line 29 “(4.55 and 1.92 mg/L, respectively)” please remove the second respectively

Answer: the ‘respectively’ inside brackets was removed

Comment Line 40 – please correct “address” to “addressed”

Answer: address was corrected in addressed

Comment: line 62 – the abstract and the article should be considered separate texts, please define the abbreviation in the text also (CA). line 64 – If CA is already abbreviated then use the abbreviated form after

Answer: As suggested by the reviewer the abbreviation of citric acid (CA) was defined in the introduction and used through out the text

Comment: Line 89 – please put a space between the number and ℃

Answer: we added a space between the number and °C

Comment: line 92 –? - the author wanted to specify the spore suspension adjustment? 106?

Answer: as suggested by the reviewer the spore concentration was specified: 5×105 conidia/mL

Comment: line 93 – RH – please define the abbreviation

Answer: we added in the text ‘relative humidity (RH)’

Comment: Line 102 - Please introduce the country where the instruments (pH-meter, LCQ Fleet ion trap spectrometer, etc) were manufactured.

Answer: as suggested by the reviewer we add the country of each instrument as followed: Microbox, Micropoli, Cesano Boscone, IT; model 720, Binder, Tuttlingen, DE; pH-meter (Hanna Instruments, Woonsocket, RI, USA; LCQ Fleet ion trap spectrometer (Thermo Fisher Scientific, Waltham, MA, USA); commercial CA (CA Com) (Sigma-Aldrich, ,St. Louis, MO, USA)

Comment: line 242, 258 - please correct to italics (in the whole manuscript).

Answer: we read through out the text and all the fungal species were correct to italics

Reviewer 2 Report

I am pleased to read this manuscript. The authors prepared the article very well. The introduction is a bit lengthy but provides comprehensive background information about the issue based on relevant references.
The research methodology is presented in a clear, legible and, above all, understandable way.
The only drawback is that there is no separate subchapter: Statisticalanalysis. You have to look for information with the help of which program has analyzed the statistical correlations between the groups. Therefore, the authors are kindly asked to take this suggestion into account and to provide references.
Another drawback is the lack of a section: Chemicals, which should include all necessary data on the origin of reagents and reference substances, along with information about the producer and country of origin. Similarly, it would be worth entering information about the analytical equipment along with a description of the entire analytical process: phase, gas pressure, temperature gradient during the analysis. Nevertheless, it is important to indicate how the recovered metals were identified from the starting samples tested.
I believe the chromatogram in the materials. additional ones should be included in the main text and, of course, properly described. It is an important element of this type of articles as it documents the experiments carried out and indicates the effective aspects of the selection of the analysis factors. It seems to me that there should be more of these chromatograms, depending on the analyzed ones. samples. They do not have to be large chromatogram formats, but can be a single graph. Similar to that shown in Figure 2 relating to citric acid.
Some doubts also apply to the summary. Authors are asked to indicate specific conclusions for the practice.

Author Response

Here, we answer point by point to comments raised by reviewers to the manuscript entitled "Food waste-assisted metal extraction from printed circuit boards: the Aspergillus niger’s route”

We have revised the manuscript in line with the following points raised by the Reviewer.

Comment: The only drawback is that there is no separate subchapter: Statistical analysis. You have to look for information with the help of which program has analyzed the statistical correlations between the groups. Therefore, the authors are kindly asked to take this suggestion into account provide references.

Answer: As suggested by the reviewer we add a specific section entitled “statistical analysis” ‘The statistical analysis were performed by Microsoft Excel (2019). The statistical correlation between groups was analysed by Student's t test. The p-value are indicated with asterisks *p < 0.05; **p < 0.01; ***p < 0.001. Differences were considered significant at p values of less than 0.05.’

Comment: Another drawback is the lack of a section: Chemicals, which should include all necessary data on the origin of reagents and reference substances, along with information about the producer and country of origin. Similarly, it would be worth entering information about the analytical equipment along with a description of the entire analytical process: phase, gas pressure, temperature gradient during the analysis.

Answer: As one of the purpose of this text is the definition of a green process, we used the strictly necessary reagents. As suggested by the reviewer, we added the Chemicals section as follow ‘2.1 Chemicals

Potato dextrose agar, methanol, citric acid, sodium hydroxide, phenolphthalein, gallium standard solution were all purchased from Sigma-Aldrich, St. Louis, MO, USA.’

Moreover, details concerning the producer and country of origin of used materials and equipment were specified along the text.

The experimental conditions of the analytical process were better specified. More precisely, following the kind suggestion of the reviewer , we described in detail the direct infusion ESI condition by adding capillary voltage, tube lens and gas flow rates parameters in the Materials and Methods section. For clarity we added these sentences ‘The instrument was set in negative ionization mode with a 5.0 kV spray voltage, 225 °C -1.0 V capillary voltage and -40 V tube lens values. Gas flow rates (arb): sheat 10, aux 0, sweep 0. Infusion flow rate was set to 5 mL/min.

Comment: Nevertheless, it is important to indicate how the recovered metals were identified from the starting samples tested.

Answer: The focus of this paper is the idea that a food waste can assist the metal extraction from a WEEE without any physical and chemical pre-treatment, among others the PCB grinding. The use of the entire WPCB, despite being engaging, entails different technical problems such as the quantification of metals in the WPCB and accessibility of metal contained in the different layer. Furthermore, in this study we used ‘Mouse’ fallen into disuse and collected by non-profit local organization “Cauto”. Even if, among WEEE, it was very difficult to find many identical PCB, this scenario is representative of the real waste management. Moreover, as the metal content of the PCB vary among boards (Bizzo et al., 2014 - DOI: 10.3390/ma7064555), metal extraction studies required destructive procedure such as disassembling, milling, chemical and thermal treatments to obtain a homogenous sample. In this study, these energy-consuming pre-treatments were eliminated to explore the possibility to submerge an entire WPCB in CA produced by SSF on food waste without any nutrient supplement. For all these reasons and for the aim of this paper, the quantification of metals in the PCB do not add more data related to the use of SSF-derived organic acids to metal leaching. The experimental design represents a new concept in waste management: A type of waste that is used to recycle another type of waste.

We keep working on this route in view of publishing more detailed data on it such as metal recovery efficiency and microscopic description of the process even introducing treatments of the PCB just to gather the most information. Currently, as an example we reported images at stereomicroscope of the same experiment after six months (Figures A and B).

Comment: I believe the chromatogram in the materials additional ones should be included in the main text and, of course, properly described. It is an important element of this type of articles as it documents the experiments carried out and indicates the effective aspects of the selection of the analysis factors. It seems to me that there should be more of these chromatograms, depending on the analyzed ones. samples. They do not have to be large chromatogram formats, but can be a single graph. Similar to that shown in Figure 2 relating to citric acid.
Some doubts also apply to the summary. Authors are asked to indicate specific conclusions for the practice.
Answer: As suggested by the reviewer the ESI-MS spectra of the negative control (previous online resource) was added in the figure 2. Moreover, we included four full-size ESI-MS spectra as online resource to allow the reader to better appreciate such results.

Please see the attachment for the figures

Round 2

Reviewer 1 Report

In the revised version of the manuscript, the authors fulfilled most of the requested requirements and increased the quality of the manuscript. Some minor comments: line 103 correct “processes” to “process”, and the term “Solid-State Fermentation (SSF)” should be defined at first use, and used the abbreviated form afterward. After these minor revisions, the article can be accepted for publication.

Author Response

We have revised the manuscript in line with the following points raised by the Reviewer.

Comment: some minor comments: line 103 correct “processes” to “process”, and the term “Solid-State Fermentation (SSF)” should be defined at first use, and used the abbreviated form afterward. After these minor revisions, the article can be accepted for publication.

Answer: Processes was correct into process.

The acronym SSF was defined at the end of introduction section and then used through out the text

Reviewer 2 Report

The presented version of the manuscript is significantly improved. The authors responded and took into account the questionable suggestions. Therefore, I believe that it is ready for publication in its current form. 

Author Response

Dear reviewer,

thanks to the prompt revision of our manuscript